# Limited Added Diagnostic Value of Whole Genome Sequencing in Genetic Testing of Inherited Retinal Diseases in a Swiss Patient Cohort

**DOI:** 10.3390/ijms25126540

**Published:** 2024-06-13

**Authors:** Jordi Maggi, Samuel Koller, Silke Feil, Ruxandra Bachmann-Gagescu, Christina Gerth-Kahlert, Wolfgang Berger

**Affiliations:** 1Institute of Medical Molecular Genetics, University of Zurich, 8952 Schlieren, Switzerland; maggi@medmolgen.uzh.ch (J.M.); koller@medmolgen.uzh.ch (S.K.); feil@medmolgen.uzh.ch (S.F.); 2Institute of Medical Genetics, University of Zurich, 8952 Schlieren, Switzerland; ruxandra.bachmann@mls.uzh.ch; 3Department of Ophthalmology, University Hospital Zurich and University of Zurich, 8091 Zurich, Switzerland; christina.gerth-kahlert@usz.ch; 4Zurich Center for Integrative Human Physiology (ZIHP), University of Zurich, 8057 Zurich, Switzerland; 5Neuroscience Center Zurich (ZNZ), University and ETH Zurich, 8057 Zurich, Switzerland

**Keywords:** whole genome sequencing (WGS), whole exome sequencing (WES), added diagnostic value, diagnostic yield, genetic testing, molecular diagnostics, inherited retinal dystrophy (IRD), structural variants (SVs), copy number variants (CNVs), deep-intronic variants

## Abstract

The purpose of this study was to assess the added diagnostic value of whole genome sequencing (WGS) for patients with inherited retinal diseases (IRDs) who remained undiagnosed after whole exome sequencing (WES). WGS was performed for index patients in 66 families. The datasets were analyzed according to GATK’s guidelines. Additionally, DeepVariant was complemented by GATK’s workflow, and a novel structural variant pipeline was developed. Overall, a molecular diagnosis was established in 19/66 (28.8%) index patients. Pathogenic deletions and one deep-intronic variant contributed to the diagnostic yield in 4/19 and 1/19 index patients, respectively. The remaining diagnoses (14/19) were attributed to exonic variants that were missed during WES analysis due to bioinformatic limitations, newly described loci, or unclear pathogenicity. The added diagnostic value of WGS equals 5/66 (9.6%) for our cohort, which is comparable to previous studies. This figure would decrease further to 1/66 (1.5%) with a standardized and reliable copy number variant workflow during WES analysis. Given the higher costs and limited added value, the implementation of WGS as a first-tier assay for inherited eye disorders in a diagnostic laboratory remains untimely. Instead, progress in bioinformatic tools and communication between diagnostic and clinical teams have the potential to ameliorate diagnostic yields.

## 1. Introduction

Inherited retinal dystrophies (IRDs) are a group of genetic disorders that affect the retina, with an estimated worldwide prevalence of 1:3450 [1]. These conditions are characterized by the degeneration or dysfunction of retinal cells, particularly rod and cone photoreceptors, which are essential for vision. The genetic etiology of IRDs is highly heterogeneous, with variants in more than 300 associated loci, and complex, with specific variants described to lead to different clinical phenotypes [2]. For these reasons, next-generation sequencing (NGS) has been widely adopted in molecular diagnostic workflows for this group of disorders [3].

Establishing a molecular diagnosis in patients affected by IRDs is crucial to confirming or improving the clinical diagnosis and informing the prognosis. While few therapies currently exist, a molecular diagnosis could enable personalized management and treatment strategies. It is a prerequisite for patients to access clinical trials or approved gene therapies for the condition. Additionally, genetic counseling can be facilitated, and valuable information can be provided during family planning for patients of childbearing age. Finally, it plays a crucial role in advancing research and understanding the underlying pathophysiologic mechanisms of IRDs, which could lead to the identification of novel, actionable therapeutic targets.

NGS has enabled the molecular diagnostics of inherited disorders, including neuropathies [4], ichthyoses [5], cardiomyopathies [6], epileptic encephalopathies [7], and neuromuscular disorders [8], to significantly improve in quality and yield. The application of NGS in diagnosing IRDs has allowed for a more comprehensive and efficient analysis of the genetic landscape underlying these conditions. Strategies including targeted sequencing (TS), whole exome sequencing (WES), and whole genome sequencing (WGS) have been used to diagnose IRDs, resulting in diagnostic yields of 50–70% [3,9].

This study aims to assess the added diagnostic value of WGS in a cohort of eye disease patients who were not diagnosed molecularly after WES and long-range PCR-based whole-gene sequencing [10]. Nineteen out of sixty-six (28.8%) index patients included in the study received a molecular diagnosis by WGS analysis. Retrospectively, only one family (1.5%) could not have been diagnosed by WES analysis. The pathogenic variants in the remaining newly diagnosed patients (18/19) were missed during the original analysis due to limitations in bioinformatic tools, newly associated loci, or their unknown significance at the time of analysis.

## 2. Results

WES analysis is being performed in our institute as a standard procedure for the genetic testing of index patients with IRDs. Between 2014 and 2021, 34.1% of index patients remained molecularly undiagnosed after WES analysis. To evaluate the added diagnostic value of WGS, we selected 66 index patients from the molecularly undiagnosed cohort based on (1) DNA sample quality and quantity, (2) family members’ availability, (3) referring medical center, and (4) date of referral. Table 1 summarizes the demographic data of the selected cohort.

### 2.1. Short Variants

Filtering and prioritization of short variants (single nucleotide variants and indels shorter than 50 bp) identified causative sequence alterations in 15/66 patients (22.7%) (Table 2). The majority of these variants (20/21) are located within or adjacent to coding regions and could have been identified with WES. They were missed during previous exome analyses due to either bioinformatics, subsequent locus-phenotype association, or insufficient evidence of pathogenicity.

During first-tier analysis, the small deletions in families 6 and 8 were not called by the bioinformatic pipeline performed as it was based around GATK’s variant caller, *UnifiedGenotyper*, which has been replaced by *HaplotypeCaller* by GATK since v4.0.0.0. The causative variants in families 10, 12, and 15 were not identified because the locus had not yet been associated with IRDs at the time of the first-tier analysis. Specifically, family 10 was first sequenced and analyzed in 2015, while variants in that gene (*SCAPER*) were described to lead to syndromic or non-syndromic retinitis pigmentosa in 2017 and 2019, respectively [21,22]. Similarly, the genes (*ACO2* and *CLN5*) responsible for the phenotype in families 12 and 15 were first described in 2021 [18,23], and the analyses were performed in 2017 and 2019, respectively.

The likely causative variants in families 1, 2, 3, 4, 7, 9, 11, 13, and 14 were detected previously by WES but were not considered to be the cause of disease due to atypical genotype–phenotype associations, incomplete or incorrect family history, or insufficient evidence of pathogenicity. In particular, families 1 and 3 carry a highly penetrant *CFH* missense variant (NM_000186.3:c.3628C>T, p.(Arg1210Cys)) [11] that was dismissed initially, yet was re-evaluated as likely causative in these families as no other likely pathogenic variant was identified in the WGS dataset. Interestingly, the index of family 1 carries the variant heterozygously, while the index of family 3 is homozygous for the variant and has been referred for genetic testing at a younger age (58 versus 69 years old). The heterozygous *CHM* variant (NM_000390.3:c.1413G>C, p.(Gln471His)) in family 4 was not considered causative during the previous analysis as the index patient is female and family history was unclear. A more detailed family history suggested dominant inheritance or atypical X-linked inheritance with incomplete penetrance or variable expressivity. Female carriers of pathogenic variants in *CHM*, a gene located on the X chromosome, have been reported to present variable clinical manifestations, which are usually milder than the symptoms of affected males [24]. Analogously, the compound heterozygous *PANK2* variants (NM_024960.4:c.395G>T, p.(Cys132Phe) and NM_024960.4:c.688G>A, p.(Gly230Arg)) in family 9 were initially filtered out as the clinical phenotype reported in the family was non-syndromic retinitis pigmentosa, which does not fit with the phenotypes described for pathogenic variants in this locus [25]. However, feedback from the clinicians indicated that the clinical manifestations of the patient extended to other neurological symptoms, which led us to re-evaluate these variants. This emphasizes the relevance of clinical information as a prerequisite for variant interpretation.

The heterozygous *IMPG2* variant (NM_016247.3:c.3423-7_3423-4del) in family 2 was originally excluded as being disease-causing for a dominantly inherited disorder due to its relatively high frequency in gnomAD. Subsequent findings have identified this variant in autosomal recessive retinitis pigmentosa patients, as well as in a patient with autosomal dominant unilateral vitelliform macular dystrophy [12,26,27,28]. Similarly, the heterozygous *HK1* variant (NM_000188.2:c.2539G>A, p.(Glu847Lys)) in family 14 was dismissed initially due to its frequency on gnomAD and unclear family history. Segregation issues were the main reasons for the exclusion of the variants in families 4, 7, and 11. However, after obtaining more detailed clinical information on family members, these variants were re-assessed as likely causative.

The heterozygous *OVOL2* 5′UTR variant (NM_021220.2:c.-61G>A) in family 13 has recently been published and described functionally to increase promoter activity [19]. The heterozygous *CRB1* deep-intronic variant (NM_201253.2:c.3879-1203C>G, p.(Trp1293_Cys1294insPhe*)) in family 5 was described to cause the inclusion of an out-of-frame pseudoexon [15]. Pangolin [29] and SpliceAI [30] predict pseudoexon inclusion, or acceptor and donor gains, with scores of 0.4, 0.35, and 0.29, respectively. The scores may not be sufficient for reporting the variant in a molecular diagnostic setting. Functional studies for *OVOL2* and the deep-intronic *CRB1* variant were crucial for diagnosing these families.

The only likely causative short variant found in this study that could only be identified by WGS is the deep-intronic *CRB1* variant in family 5 (NM_201253.2:c.3879-1203C>G, p.(Trp1293_Cys1294insPhe*)).

### 2.2. Structural Variants

Structural variant analysis revealed likely causative deletions encompassing coding regions of IRD-associated genes in four families (6.1%) (Table 3). These deletions include coding regions and have the potential to be identified with WES. However, identification of copy number variants (CNVs) from WES data is challenging due to variable coverage within target regions. The implementation of cn.Mops (https://www.bioinf.jku.at/software/cnmops/cnmops.html, accessed on 10 January 2020) [31] and the SeqNext module of Sequence Pilot version 5.0 (JSI Medical Systems GmbH, Ettenheim, Germany) in our WES analysis pipeline have been instrumental in the detection of large CNVs [32,33]. Nevertheless, detecting CNVs affecting a few exons from WES data remains demanding, partially due to the high number of false positives [34,35]. Therefore, these cases are considered to contribute to the added diagnostic value of genome sequencing. Retrospectively, these CNVs could be detected by Sequence Pilot (Appendix A).

### 2.3. Diagnostic Yield and Added Diagnostic Value

Analysis of genome sequencing datasets led to the identification of a molecular diagnosis in 19/66 families (28.8%) that were undiagnosed after WES analysis.

We define the added diagnostic value of WGS as the molecular diagnoses made possible by WGS by the identification of pathogenic structural variants (which are unreliably detected by WES analysis) and short variants in noncoding regions of the genome (such as those located deep-intronically or in regulatory regions, for example). To calculate this, families diagnosed with variants detectable by WES (14 families) were excluded, resulting in an added diagnostic value of 9.6% (5/52 families). Similar studies revealed comparable added diagnostic value for WGS for inherited retinal dystrophies, ranging from 1.6% to 18.5% (Table 3).

In this study, we identified only one patient (family 5) carrying a published disease-causing deep-intronic variant. Genome sequencing contributed mainly to diagnostic yield by allowing the detection of CNVs in four families. Similarly, previous studies reported 3–6× more families diagnosed by the detection of structural variants than intronic or regulatory short variants (Table 4).

Candidate variants in IRD-associated loci were identified in most undiagnosed families (41/47) (Appendix A), but additional data, such as functional assays or samples from family members, would be needed to conclusively interpret their pathogenicity.

**Table 3 ijms-25-06540-t003:** Newly diagnosed families by structural variants. ^1^ Variant gNomen: NC_000004.11:g.47931964_47946798del; ^2^ Variant gNomen: NC_000006.11:g.66151762_66620415del; ^3^ Variant gNomen: NC_000016.9:g.28497286_28498251del; ^4^ Variant gNomen: NC_000010.10:g.94348029_94365528delinsGCATGAGCCTGAGATCAAGG. * Variants in regions covered exclusively by WGS have only been sequenced for patients affected by inherited retinal dystrophies included in this study; therefore, the overall in-house database minor allele frequency and the IRD-only minor allele frequency have the same value. The lowest in-house frequency is 0.76% (1/132 alleles).

Fam.	Clinical Phen.	Age at Ref.	Gene	Variant (cNomen)	Size (kb)	Affected Exon(s)	Zyg.	gnomAD All (%)	gnomAD Max (%)	In-House (%)	IRD (%)	ACMG	HGMD	ClinVar	Ref.	Seg.
16	RP	16	*CNGA1*	NM_001142564.1:c.507-1368_*6475del ^1^	14.8	ex. 6–10	Hom.	0	0	1.52 *	1.52 *	VUS	-	-	TS	Y
17	RP	26	*EYS*	NM_001142800.1:c.-203835_863-36502del ^2^ NM_001142800.1:c.8543T>G (p.(Ile2848Ser))	468.7	ex. 1–5	Het.	0	0	0.76 *	0.76 *	LP	-	-	TS	NA
							Het.	0	0	0.063	0.11	VUS	-	-	TS	
18	CRD	11	*CLN3*	NM_001042432.1:c.461-280_677+382del ^3^ NM_001042432.1:c.883G>A (p.(Glu295Lys))	0.97	ex. 8–9	Het.	0.125	0.556	1.52 *	1.52 *	P	P	-	[36]	Y
							Het.	0.003	0.014	0.063	0.11	P	P	P	[37]	
19	CRD	34	*KIF11*	NM_004523.4:c.-5104_78-494delins GCATGAGCCTGAGATCAAGG ^4^	17.5	ex. 1	Het.	0	0	0.76 *	0.76 *	LP	-	-	TS	Y

Abbreviations: Fam., family; Clinical phen., clinical phenotype; Age at ref., age at referral; gNomen, HGVS genomic-level nucleotide change nomenclature; cNomen, HGVS cDNA-level nucleotide change nomenclature; Zyg., zygosity; gnomAD all (%), genome aggregation database overall minor allele frequency in percentage; gnomAD max (%), genome aggregation database highest minor allele frequency in percentage; in-house (%), in-house database overall minor allele frequency in percentage; IRD (%), in-house database inherited retinal dystrophy patients-only minor allele frequency in percentage; ACMG, American College of Medical Genetics and Genomics guidelines; HGMD, Human Gene Mutation Database; Ref., reference; Seg., segregation; RP, retinitis pigmentosa; CRD, cone-rod dystrophy; Hom., homozygous; Het., heterozygous; VUS, variant of unknown significance; P, pathogenic; LP, likely pathogenic; TS, this study; Y, the variant segregates disease within the family; NA, not available.

**Table 4 ijms-25-06540-t004:** Diagnostic yield and genome-sequencing added diagnostic value for this and similar studies. ^1^ Number of families diagnosed by detecting structural and/or intronic and regulatory short variants. ^2^ Number of families diagnosed due to this type of variant. ^3^ This figure does not include noncanonical splice variants that are detectable by WES.

Study	Country	Year	Fam.	Cohort	Diag. Yield Overall (%)	Added Diag. Value (%) ^1^	Structural Variant ^2^	Int. or Reg. Variants ^2^	Splicing Assay
Ellingford et al. [38]	World	2016	562	IRD	52.0	18.5 (5/27)	5	0 ^3^	No
Carss et al. [39]	UK	2017	722	IRD	56.0	9.6 (34/355)	28	6 ^3^	No
Numa et al. [40]	Japan	2020	220	RP	44.5	1.6 (2/124)	2	0	No
Weisschuh et al. [41]	Germany	2023	968	IRD/ION	57.3	15.7 (77/490)	59	20	RNA-seq
Liu et al. [42]	China	2024	271	IRD	-	12.5 (34/271)	29	5	Minigene
This study	Switzerland	2024	66	IRD/MED	28.8	9.6 (5/52)	4	1	No

Abbreviations: Fam., number of families; Diag., diagnostic; IRD, inherited retinal dystrophy; RP, retinitis pigmentosa; ION, inherited optic neuropathy; MED, mixed eye disorders.

## 3. Discussion

Most diagnostic laboratories employ TS or WES for the analysis of patients affected by IRDs, which are characterized by high genetic and clinical variability [3]. Diagnostic yield rates for IRDs vary greatly based on cohort composition; for example, cohorts composed mainly of patients affected by RP have lower diagnostic yields [3]. In this study, we report an overall diagnostic yield of 28.8% in a cohort composed of mixed IRD and eye disorders patients that remained undiagnosed after previous exome analyses. It is important to note that the cohort is composed of negative cases (no diagnosis after WES analysis), which explains why the diagnostic yield is lower than usual. In previous studies, we reported yields of 64% after WES and 67% after long-range PCR-based whole-gene sequencing combined with WES for unselected IRD cohorts [10,43].

WGS analysis allows for additional diagnoses owing to the ability to detect deep-intronic variants and to reliably call structural variants [38,39,41,42]. As shown in Table 4, previous studies report that between 9.6 and 18.5% of the cohort undiagnosed after WES analysis has been molecularly diagnosed by employing WGS. In line with these studies, we established an overall added diagnostic value of 9.6% in our cohort.

The remaining new diagnoses were allowed by the employment of other bioinformatic tools, the description of new locus-phenotype associations, and additional clinical and family history data. Therefore, repeating exome analysis for undiagnosed families when updating bioinformatic workflows and gene lists can be very useful. Moreover, close collaboration with the referring ophthalmologist can prove beneficial during variant interpretation.

### 3.1. Undiagnosed Families

Forty-seven families (71.2%) remain without a definite molecular diagnosis after WGS. Of these, forty-one families carry interesting variants in IRD-associated loci, which represent candidates for pathogenic sequence alterations (Appendix A).

In family 53, a possibly pathogenic variant in *IMPDH1* (NM_000883.3:c.130G>C, p.(Gly44Arg)) has been identified. The limited clinical data and the lack of family member samples and anamnesis restricted our ability to evaluate this variant further. Conversely, variants in *CNGB1* and *MDPZ* in family 57 were excluded as being disease-causing because the family history is indicative of a dominant inheritance mode.

Families 22, 24, 26, 39, 40, 49, and 58 carry a coding and a deep-intronic variant in genes associated with a recessive form of IRD. Interestingly, families 26 and 58 have the same coding variant (NM_000350.2:c.3755A>T, p.(Glu1252Val)) and the same deep-intronic variant (NM_000350.2:c.3863-1094T>A) in *ABCA4*. Family 26, however, carries an additional rare deep-intronic variant (NM_000350.2:c.3863-1203G>T) in cis with c.3863-1094T>A. Similarly, variants that may affect splicing were detected in genes associated with dominant forms of IRD in families 23, 28, 31, and 42. Splicing assays for these candidate variants would be necessary to further evaluate them.

Rare variants in potential regulatory regions of genes associated with IRDs were identified in families 21, 43, and 54. The variant in family 54 (NM_021620.2:c.-4096T>C) is located 4096 nucleotides upstream of the transcription start site of the gene *PRDM13*. Several variants located between 14,005 and 13,871 or between 8128 and 8107 nucleotides upstream of *PRDM13* have been described to lead to North Carolina macular dystrophy (OMIM 136550) [44,45,46]. Functional assays such as luciferase assays could help re-evaluate the pathogenicity of these variants.

Several families carry variants in genes encoding proteins known to directly interact with each other. Variants in *FZD4* and *LRP5* were found in families 27 and 36; digenic inheritance for exudative vitreoretinopathy (EVR) involving variants in these genes has been reported previously [47,48]. Similarly, digenic inheritance for oculocutaneous albinism (OCA) with variants in *TYR* and *OCA2* has been suggested previously [49,50]. Variants in these genes have been discovered in family 48. Finally, two variants in genes *GUCY2D* and *RD3* were identified in families 20 and 21; the proteins encoded by these genes are known to interact directly [51]. The limited number of family members available for analysis prevented us from further evaluating the findings in these families.

Variants in genes associated with a dominantly inherited eye disorder with an allele count higher than 1 in the gnomAD database were considered likely benign (i.e., families 25, 28, 56, and 60), except for late-onset phenotypes (i.e., families 1 and 14). Segregation within the family further excluded dominant candidate variants in families 25, 28, 52, and 60. Interestingly, the index patient of family 25 carries a de novo variant in *PITPNM3* (NM_031220.3:c.2000G>A, p.(Gly667Glu)), which excludes it from consideration as the causative variant because the affected mother does not carry it. Family 38 carries a *PIKFYVE* variant (NM_015040.3:c.4511G>A, p.(Trp1504*)) that has been described in a patient affected by fleck corneal dystrophy [52]. The ophthalmological examination of the index patient of family 38, however, did not reveal evident abnormalities in the cornea.

### 3.2. Structural Variants

In previous studies, structural variants contributed between 76.6 and 100% to the added diagnostic value of WGS, or between 3.9 and 8.9% of the entire cohort [38,39,41,42]. In this study, structural variants contributed to 80% (4/5) of the additional diagnoses. The integration of CNV analysis in WES bioinformatic pipelines is challenging due to the plethora of tool options, the low concordance in calls from different tools, the relatively high false-positive calls, and the low sensitivity [34]. Despite these limitations, CNV analysis from WES data has led to an increase in diagnostic yield in mixed cohorts [53,54], as well as in IRD cohorts [55,56]. The implementation of CNV analysis in the exome bioinformatic pipeline may, therefore, reduce the added diagnostic value of WGS. The integration of multiple CNV calling tools in the pipeline could improve specificity and sensitivity. Recent efforts to focus on alternative CNV-related contaminating data (off-target reads) from WES datasets have shown improved sensitivity and specificity [57].

### 3.3. Deep-Intronic Variants

Deep-intronic variants constitute a prerogative of WGS, or targeted whole-gene sequencing, as they are not covered by WES. So far, a relatively small fraction of the additional diagnoses (up to 26.0% of the added diagnostic value, or 2.1% of the entire cohort, Table 4) have been allowed by the detection of this type of variant. Comparatively, we report a single deep-intronic variant contributing to the diagnostic yield in our cohort (20% of the added diagnostic value, Table 4). However, splicing assays for variants in families 22, 23, 24, 26, 28, 31, 39, 40, 42, 49, and 58 may boost this figure significantly.

The main bottleneck for the added diagnostic value of WGS lies in the lack of appropriate prediction tools to interpret these variants in a diagnostic setting. The deep-intronic variants contributing to diagnostic yields in the studies included in Table 4 have been described to affect splicing by a functional assay. Pangolin and spliceAI [29,30] have proved useful in prioritizing candidate deep-intronic splicing variants; they do not, however, provide enough evidence to classify a deep-intronic variant as pathogenic or likely pathogenic without a functional assay. Therefore, the contribution of deep-intronic variants is likely underestimated.

Liu et al. utilized minigene assays to support the predicted splicing-altering effects of candidate deep-intronic and noncanonical splice-site variants [42]. Weisschuh et al. and Nash et al. employed RNA-seq data from different patient-derived tissues (whole blood and skin biopsy, respectively) to verify aberrant splicing due to deep-intronic variants in *SDCCAG8* and *IQCB1*, respectively [41,58]. Complementary diagnostic transcriptome analysis of accessible tissues in combination with WGS has the potential to overcome the challenges of candidate splicing-altering variant interpretation [59]. The utility of this strategy in the field of IRD diagnostics is limited by the inaccessibility of relevant tissues. Instead, more accessible tissues, such as whole blood or a skin biopsy, have been used so far [41,58]. Weisschuh et al. reported that only a fraction of the most frequently mutated genes (5/28) in their cohort are stably expressed in blood [41].

### 3.4. Limitations

The relatively small cohort size limits our ability to draw further and more general conclusions. Additionally, the limited availability of samples from additional family members for testing and the lack of detailed clinical information restricted our ability to interpret variants of unknown significance.

## 4. Materials and Methods

### 4.1. Cohort Selection

Index patients (*n* = 66) were referred to us for genetic testing between 2002 and 2020 from large specialized medical/ophthalmologic centers in Switzerland and abroad. Blood samples were collected from index patients and available family members. Written informed consent was obtained from all probands and family members included in this study, which was conducted in accordance with the 2013 Declaration of Helsinki. The probands included in this study have undergone WES analysis previously (between 2014 and 2021) but have remained molecularly undiagnosed. The probands are affected by a form of IRD (macular dystrophy *n* = 26, retinitis pigmentosa *n* = 17, cone-rod dystrophy *n* = 6, exudative vitreoretinopathy *n* = 6, cone dystrophy *n* = 2, choroideremia *n* = 2, Usher syndrome *n* = 1, Leber congenital amaurosis *n* = 1), optic atrophy (*n* = 3), Wagner syndrome (*n* = 1), or corneal dystrophy (*n* = 1). A cohort overview is included in Table 1.

### 4.2. Genetic Testing

Genomic DNA was extracted from whole blood in duplicate with the automated Chemagic MSM I system according to the manufacturer’s specifications (PerkinElmer Chemagen Technologie GmbH, Baesweiler, Germany). Whole genome sequencing was performed on 66 DNA samples. One microgram of gDNA was sheared on a Covaris M220 (Covaris, Woburn, MA, USA) in a 50 µL Covaris Adaptive Focused Acoustic (AFA) microtube (Covaris, Woburn, MA, USA) to a target size of 350 bp. Subsequently, fragmented gDNAs were processed with the TruSeq Nano DNA PCR-Free kit according to the manufacturer’s instructions (Illumina, San Diego, CA, USA). Finally, 64 of these libraries were diluted, pooled, and sequenced at the Functional Genomics Center Zurich (FGCZ) on a NovaSeq 6000 (Illumina, San Diego, CA, USA) to a target coverage of 50X. The other two libraries were sequenced to a target coverage of 30X either in-house on a NextSeq500 instrument (Illumina, San Diego, CA, USA) or at the Cologne Center for Genomics (CCG) on a HiSeq2500 instrument (Illumina, San Diego, CA, USA).

### 4.3. Sequencing Data Analysis

Illumina sequencing data were aligned to the human reference genome hg19 according to GATK (v4.2.6.1) Best Practices [60] for single-sample short germline variants discovery (https://gatk.broadinstitute.org/hc/en-us/articles/360035535932-Germline-short-variant-discovery-SNPs-Indels, accessed on 16 August 2021) with the Burrows–Wheeler Aligner (BWA) (v0.7.17) [61]. Additionally, DeepVariant (v1.3.0) [62] was utilized to create a supplementary variant call set. Only variants that were called by both tools (GATK HaplotypeCaller [63] and DeepVariant) were kept. The resulting Variant Call Format (VCF) files were annotated with the Nirvana (https://illumina.github.io/NirvanaDocumentation/, accessed on 6 October 2021) annotator.

For structural variant identification, we developed an analysis pipeline that integrates six structural and copy number variant callers: Lumpy (v0.3.1) [64], Manta (v1.6.0) [65], CNVpytor (v1.2.1) [66], Delly2 (v1.0.3) [67], GRIDSS (v2.11.1) [68], and WHAM (v1.8.0.1.2017.05.03) [69]. The merged VCFs were annotated with the Nirvana annotator.

The entire analysis pipelines, as well as the filtering and conversion scripts, are available on github (https://github.com/jordimaggi/WGS_anaylsis_workflow).

### 4.4. Variants Database

Annotated WGS (66 files) and WES VCFs (925 files) produced over the years were gathered to build an in-house rare (gnomAD “all” < 1%) variant database. The database reports disease group-specific (e.g., IRD-specific) variant frequencies that could aid in the interpretation of recurrent variants. Only index patients were included in the calculation of variant frequencies. Cohort-specific and disease group-specific variant frequency data were extracted and converted to be included as a custom annotation field during subsequent annotations with Nirvana.

### 4.5. Variant Interpretation

Nirvana-annotated JSON files were converted to a tabular format and filtered with an eye disorders loci list (Appendix A) using a Python script in a stepwise manner. First, only variants with an overall frequency below 1% on gnomAD [70] within loci most frequently associated with the clinical phenotype reported (Appendix A) were analyzed. If no molecular diagnosis was found, rare variants (gnomAD all < 1%) affecting other eye disease-associated loci were considered. Additionally, intergenic regions that have been associated with IRDs were investigated (Appendix A).

Variant prioritization was based on gnomAD frequencies [70], ClinVar entries [71], phyloP score [72], CADD v1.6 score [73], spliceAI scores [30], primateAI score [74], revel score [75], sift prediction [76], polyPhen prediction [77], family history, and in-house frequencies. The Human Gene Mutation Database (HGMD) [78] was queried for previous reports of pathogenicity for candidate variants. The Franklin platform (https://franklin.genoox.com/clinical-db/home, accessed on 20 April 2024) was used to automatically verify 17/28 criteria from the American College of Medical Genetics and Genomics (ACMG) guidelines to classify candidate variants [79]. When available, segregation data were manually curated to re-assess variant classification.

### 4.6. Segregation Analysis

Segregation analysis for candidate variants was performed if family members were available. For this purpose, initial PCR amplification and Sanger sequencing were performed as previously described [32]. Alternatively, long-range PCR was performed as previously described [10].

## 5. Conclusions

Given the limited added diagnostic value of WGS when compared to WES (especially when robust CNV analysis is implemented) and the 2–3× higher associated costs [80], the application of WGS as a first-tier assay in the context of a diagnostic laboratory with limited resources remains questionable at the current point in time. As a first-tier assay, the most cost-effective approach may be TS of loci associated with IRDs or WES, including intronic and regulatory regions that have been described in the literature. Genome sequencing for undiagnosed patients may follow as a second-tier assay performed in large and specialized centers.

In a research setting, however, the focus should be on WGS combined with minigene assays or transcriptome analysis of retinal organoids or retinal pigment epithelial cells obtained from patient-derived induced pluripotent stem cells. Research in this field will help identify and interpret novel deep-intronic and regulatory variants affecting gene function, which may lead to improved bioinformatic prediction tools.

## Figures and Tables

**Table 1 ijms-25-06540-t001:** Demographic data of the probands included in the cohort. Age at referral and sex refer to the index patient.

Family	Clinical Phenotype	Sex	Age at Referral	Family History	Status	Gene
1	MD	M	69	ND	Diagnosed	*CFH*
2	MD	M	47	ND	Diagnosed	*IMPG2*
3	CACD	F	58	dominant	Diagnosed	*CFH*
4	RP	M	63	dominant	Diagnosed	*CHM*
5	RP	F	8	negative	Diagnosed	*CRB1*
6	RP	F	54	dominant	Diagnosed	*RP1*
7	EVR	F	0	negative	Diagnosed	*KIF11*
8	RP	F	8	recessive	Diagnosed	*NRL*
9	RP	M	29	negative	Diagnosed	*PANK2*
10	RP	F	31	recessive	Diagnosed	*SCAPER*
11	RP	M	50	negative	Diagnosed	*KLHL7*
12	OPA	F	43	unclear	Diagnosed	*ACO2*
13	CD	M	67	dominant	Diagnosed	*OVOL2*
14	MD	F	71	dominant	Diagnosed	*HK1*
15	COD	M	35	negative	Diagnosed	*CLN5*
16	RP	M	16	negative	Diagnosed	*CNGA1*
17	RP	M	26	negative	Diagnosed	*EYS*
18	CRD	F	11	negative	Diagnosed	*CLN3*
19	CRD	M	34	unclear	Diagnosed	*KIF11*
20	CRD	M	45	ND	Undiagnosed	
21	MD	M	43	unclear	Undiagnosed	
22	CRD	F	33	ND	Undiagnosed	
23	VMD	F	43	ND	Undiagnosed	
24	MD	F	27	negative	Undiagnosed	
25	MD	F	11	dominant	Undiagnosed	
26	MD	F	34	ND	Undiagnosed	
27	EVR	F	9	negative	Undiagnosed	
28	RP	F	31	recessive	Undiagnosed	
29	MD	F	34	dominant	Undiagnosed	
30	CRD	M	45	dominant	Undiagnosed	
31	STGD	M	49	ND	Undiagnosed	
32	RP	M	33	negative	Undiagnosed	
33	RP	M	34	dominant	Undiagnosed	
34	MD	F	34	recessive	Undiagnosed	
35	MD	M	45	ND	Undiagnosed	
36	EVR	F	6	negative	Undiagnosed	
37	RP	F	32	negative	Undiagnosed	
38	DHDD	M	43	ND	Undiagnosed	
39	STGD	M	10	negative	Undiagnosed	
40	MD	F	33	negative	Undiagnosed	
41	RP	F	43	negative	Undiagnosed	
42	MD	F	32	negative	Undiagnosed	
43	MD	M	67	ND	Undiagnosed	
44	RP	F	30	recessive	Undiagnosed	
45	MD	M	37	recessive	Undiagnosed	
46	MD	F	46	negative	Undiagnosed	
47	WGN	M	30	ND	Undiagnosed	
48	OCA	M	32	dominant	Undiagnosed	
49	RD	M	2	negative	Undiagnosed	
50	EVR	F	1	negative	Undiagnosed	
51	LHON	F	26	ND	Undiagnosed	
52	USH	F	10	negative	Undiagnosed	
53	CHM	M	58	ND	Undiagnosed	
54	MD	F	39	negative	Undiagnosed	
55	COD	M	10	recessive	Undiagnosed	
56	RP	M	61	negative	Undiagnosed	
57	MD	M	13	dominant	Undiagnosed	
58	MD	M	55	recessive	Undiagnosed	
59	RP	M	23	dominant	Undiagnosed	
60	RP	F	20	negative	Undiagnosed	
61	MD	F	58	ND	Undiagnosed	
62	CRD	F	45	ND	Undiagnosed	
63	EVR	M	1	negative	Undiagnosed	
64	EVR	M	9	dominant	Undiagnosed	
65	MD	F	10	negative	Undiagnosed	
66	OPA	M	65	ND	Undiagnosed	

Abbreviations: MD, macular dystrophy; CACD, central areolar choroidal dystrophy; RP, retinitis pigmentosa; EVR, exudative vitreoretinopathy; OPA, optic atrophy; CD, corneal dystrophy; COD, cone dystrophy; CRD, cone-rod dystrophy; VMD, vitelliform macular dystrophy; STGD, Stargardt’s disease; DHDD, Doyne honeycomb retinal dystrophy; WGN, Wagner syndrome; OCA, oculocutaneous albinism; RD, retinal dystrophy; LHON, Leber hereditary optic neuropathy; USH, Usher syndrome; CHM, choroideremia; M, male; F, female; ND, no data.

**Table 2 ijms-25-06540-t002:** Families newly diagnosed by short variants. * Variants in regions covered exclusively by WGS have only been sequenced for patients affected by inherited retinal dystrophies included in this study; therefore, the overall in-house database minor allele frequency and the IRD-only minor allele frequency have the same value. Therefore, the lowest in-house frequency equals 0.76% (1/132 alleles).

Fam.	Clinical Phen.	Age at Ref.	Gene	Variant (cNomen)	pNomen	Zyg.	gnomAD All (%)	gnomAD Max (%)	In-House (%)	IRD (%)	ACMG	HGMD	ClinVar	Ref.	Seg.
1	MD	69	*CFH*	NM_000186.3:c.3628C>T	p.(Arg1210Cys)	Het.	0.03	0.06	0.32	0.44	LP	P	P/VUS	[11]	NA
2	MD	47	*IMPG2*	NM_016247.3:c.3423-7_3423-4del	p.?	Het.	0.01	0.02	0.06	0.11	LP	P	P/LP/VUS	[12]	NA
3	CACD	58	*CFH*	NM_000186.3:c.3628C>T	p.(Arg1210Cys)	Hom.	0.03	0.06	0.32	0.44	LP	P	P/VUS	[11]	NA
4	RP	63	*CHM*	NM_000390.3:c.1413G>C	p.(Gln471His)	Het.	0	0	0.19	0.33	VUS	P	-	[13]	NA
5	RP	8	*CRB1*	NM_201253.2:c.2401A>T	p.(Lys801*)	Het.	0.006	0.028	0.06	0.11	P	P	P	[14]	Y
				NM_201253.2:c.3879-1203C>G	p.(Trp1293_Cys1294insPhe*)	Het.	0.003	0.007	0.76 *	0.76 *	VUS	P	-	[15]	
6	RP	54	*RP1*	NM_006269.1:c.2285_2289del	p.(Leu762Tyrfs*17)	Het.	0	0	0.06	0.11	P	P	P	[16]	Y
7	EVR	0	*KIF11*	NM_004523.3:c.1875+2T>A	p.?	Het.	0	0	0.08	0.12	LP	-	-	TS	Y
8	RP	8	*NRL*	NM_006177.3:c.-41_-28+23del	p.?	Hom.	0	0	0.44	0.88	VUS	-	-	TS	Y
9	RP	29	*PANK2*	NM_024960.4:c.395G>T	p.(Cys132Phe)	Het.	0.001	0.016	0.06	0.11	LP	-	-	TS	Y
				NM_024960.4:c.688G>A	p.(Gly230Arg)	Het.	0.013	0.042	0.06	0.11	P	P	P	[17]	
10	RP	31	*SCAPER*	NM_020843.2:c.334C>T	p.(Arg112*)	Hom.	0.002	0.003	0.13	0.22	P	-	-	TS	Y
11	RP	50	*KLHL7*	NM_001031710.2:c.1191_1192del	p.(Tyr398Phefs*3)	Het.	0	0	0.06	0.11	LP	-	-	TS	Y
12	OPA	43	*ACO2*	NM_001098.2:c.2006C>T	p.(Ser669Leu)	Het.	0.0004	0.001	0.06	0.11	VUS	P	VUS	[18]	NA
13	CD	67	*OVOL2*	NM_021220.2:c.-61G>A	p.?	Het.	0	0	0.06	0.11	LP	P	-	[19]	NA
14	MD	71	*HK1*	NM_000188.2:c.2539G>A	p.(Glu847Lys)	Het.	0.001	0.011	0.06	0.11	LP	P	P/LP	[20]	NA
15	COD	35	*CLN5*	NM_006493.2:c.445C>A	p.(Leu149Ile)	Hom.	0	0	0.13	0.22	VUS	-	VUS	TS	Y

Abbreviations: Fam., family; Clinical phen., clinical phenotype; Age at ref., age at referral; cNomen, Human Genome Variation Society (HGVS) cDNA-level nucleotide change nomenclature; pNomen, predicted protein-level change nomenclature; Zyg., zygosity; gnomAD all (%), genome aggregation database overall minor allele frequency in percentage; gnomAD max (%), genome aggregation database highest minor allele frequency in percentage; in-house (%), in-house database overall minor allele frequency in percentage; IRD (%), in-house database inherited retinal dystrophy patients-only minor allele frequency in percentage; ACMG, American College of Medical Genetics and Genomics guidelines; HGMD, Human Gene Mutation Database; Ref., reference; Seg., segregation; MD, macula dystrophy; CACD, central areolar choroidal dystrophy; RP, retinitis pigmentosa; EVR, exudative vitreoretinopathy; OPA, optic atrophy; CD, corneal dystrophy; Hom., homozygous; Het., heterozygous; VUS, variant of unknown significance; P, pathogenic; LP, likely pathogenic; TS, this study; Y, the variant segregates disease within the family; NA, not available.

## Data Availability

The original contributions presented in the study are included in the article/Appendix A, further inquiries can be directed to the corresponding authors.

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
