# Peer review of "Limited Added Diagnostic Value of Whole Genome Sequencing in Genetic Testing of Inherited Retinal Diseases in a Swiss Patient Cohort"

_ijms, 2024, doi:10.3390/ijms25126540_

Round 1

Reviewer 1 Report

Comments and Suggestions for Authors

The articles “Limited added diagnostic value of whole genome sequencing in genetic testing of inherited retinal diseases in a Swiss patient cohort” by Maggi et al., study the diagnostic value of whole genome sequencing for patients with inherited retinal diseases  who remained undiagnosed after whole exome sequencing of database of index patients of 66 families analyzed using GATK’s guidelines. The draft is well written and compiled.

My comments/observations to improve the manuscript are as follows:

1.      Add some text after heading, 2. Results and before 2.1.

2.      Please add, inclusion and exclusion criteria for the family.

3.      Table 1 caption is too huge and complex. Make it simple and keep some text at footer.

4.      Table 1 and 2: some raw don’t have references.

5.      Line 132-134: Please cite, However, feedback from the clinicians indicated that the clinical manifestations of the patient extended to other neurological symptoms, which led us to re-evaluate these variants. This emphasizes the relevance of clinical information as a prerequisite for variant interpretation.

6.      Table 2 caption need to relook, too long.

7.      Table 3: need uniformity, as table 1 and 2.

8.      Line 389: Cohort-, is this – is purposeful.

9.      In conclusion cost is there. Can we discuss one line of accessibility of technology. 

Reviewer 2 Report

Comments and Suggestions for Authors

1.       This was a study of 66 families evaluating the added diagnostic value of whole genome sequencing (WGS) for patients with IRDs who remained undiagnosed after whole exome sequencing (WES) and found an additional 9.6% genetic diagnoses among patients. This study is very timely and important given the numerous clinical trials and therapies being developed for patients with IRDs that depend on the genetic mutations of patients, therefore necessitating genetic testing and determination prior to enrollment in these gene-therapy based clinical trials.  This is both new and timely data. Given the limited sample size, it would be beneficial if the authors could add more details about this group of patients (detailed below) to understand how this specific group of patients is generalizable to other clinics around the world.

2.       Please provide further details about where these patients were enrolled (large tertiary care referral academic medical center vs a small private practice) and characteristics of that practice/s. Was this a single center or multiple?

3.       Please provide details on when (years) these patients were enrolled. If I understand, this was between 2014 and 2021? Please make this more explicit.

4.       Please provide information on ethical approval and consent, eg institutional review board approval of the study. While in section 4. Materials and Methods, the authors state “Written informed consent was obtained…” please also include a statement on the approval of the institutional review board and the approval number.

5.       Please add a Table or information on the demographics and characteristics of these patients, including race, ethnicity, gender, and age. This portion of Table S1 should be moved to the main manuscript and further details should be included. While the diagnoses is listed in the text on 4.1, this would be helpful to include in table format rather than strictly text format along with percentages. This is important information to provide to understand the generalizability of this data to other patient populations and clinics.

6.       Please include a flow diagram indicating: 1. how many patients were evaluated at the sites for IRDs, 2. How many patients were excluded, 3. The reason that patients were excluded.

7.       In addition to the tables giving exact details of each patient, it would be beneficial to have tables including broader percentages of characteristics to make the results section easier to read.

8.       In Table 3, the year of publication is likely less important than the years of evaluation. It would be beneficial to include that information also on the table.

9.       Please add a section discussing some limitations of this study, including the relatively small sample size, this is a single center (although not detailed in the manuscript), etc.
